REGISTERED REPORT

# Registered report: Intestinal inflammation targets cancer-inducing activity of the microbiota

Kate Eaton[1], Wanwan Yang[2], Reproducibility Project: Cancer Biology*[†]

[1]Germ Free Laboratory, University of Michigan Medical School, Ann Arbor, United States; [2]ImpeDx Diagnostics Inc., Kansas City, United States

**Abstract** The Reproducibility Project: Cancer Biology seeks to address growing concerns about reproducibility in scientific research by conducting replications of 50 papers in the field of cancer biology published between 2010 and 2012. This Registered report describes the proposed replication plan of key experiments from 'Intestinal Inflammation Targets Cancer-Inducing Activity of the Microbiota' by *Arthur et al. (2012)*, published in Science in 2012. Arthur and colleagues identified a genotoxic island in *Escherichia coli* NC101 that appeared to be responsible for causing neoplastic lesions in inflammation-induced *IL10*[−/−] mice treated with azoxymethane. The experiments that will be replicated are those reported in Figure 4 (*Arthur et al., 2012*). Arthur and colleagues inoculated *IL10*[−/−] mice with a mutated strain of *E. coli* NC101 lacking the genotoxic island, and showed that those mice suffered from fewer neoplastic lesions than mice inoculated with the wild type form of *E. coli* NC101 (Figure 4). The Reproducibility Project: Cancer Biology is a collaboration between the Center for Open Science and Science Exchange, and the results of the replications will be published by *eLife*.

*For correspondence: fraser@ scienceexchange.com

**Group author details**

[†]Reproducibility Project: Cancer Biology
See page 11

**Reviewing editor**: Cynthia L Sears, Johns Hopkins University School of Medicine, United States

## Introduction

In their 2012 Science paper, Arthur and colleagues examined the interplay between colitis and colon cancer. They identified a shift in the composition of the microbiota in *Il10*[−/−] mice, which develop chronic colitis; amongst other changes noted, *Escherichia coli* was over 100-fold more represented in *Il10*[−/−] mice than wild type. After treatment with azoxymethane (AOM), a carcinogen that induces colon cancer, germ-free *Il10*[−/−] mice mono-associated with the colitis-inducing *E. coli* NC101 strain developed invasive mucinous carcinomas, while mice mono-associated with *Enterococcus faecalis*, another colitis-inducing bacterial strain, did not. *E. coli* NC101 harbors a 54 kb *polyketide synthases* (*pks*) genotoxic island encoding several enzymes involved in the production of toxin called Colibactin. This island has been previously shown to induce DNA damage, double strand breaks and aneuploidy (*Nougayrede, 2006*; *Cuevas-Ramos et al., 2010*) and was not found in *E. faecalis* or another non-colitic *E. coli* strain, K12.

In Figure 4, Arthur et al. inoculated germ-free *IL10*[−/−] mice treated with or without AOM with either wild-type *E. coli* NC101, or with a mutant of *E. coli* NC101 lacking the *pks* island (NC1010Δ*pks*). Arthur and colleagues first confirmed that loss of the *pks* island did not impair bacterial growth (Supplemental figure 7, replicated in Protocol 1). Presence or absence of the genotoxic *pks* island had no effect on colonic inflammation in *IL10*[−/−] mice alone or treated with AOM (Figure 4A). However, at 12 and 18 weeks, mice treated with AOM and inoculated with *E. coli* NC101Δ*pks* had many fewer neoplastic lesions than mice inoculated with wild-type *E. coli* NC101 (Figure 4B). At 18 weeks, invasion (Figure 4C), tumor burden (Figure 4D) and tumor size (Figure 4E) were all reduced in mice mono-associated with *E. coli* NC101Δ*pks* as compared to NC101. Taken together, the data indicate that loss

of the *pks* genotoxic island from *E. coli* NC101 strongly reduces the incidence of colon cancer in mice with chronic colitis. These experiments are replicated in Protocol 3.

Buc et al. (2013) performed an experiment similar to Figure 3B (not included for replication in this study), wherein *Arthur et al. (2012)* examined if the *pks* genotoxic island was more prevalent in patients with colorectal cancer. Both the dataset from *Arthur et al. (2012)* and the dataset presented by *Buc et al. (2013)* support the hypothesis that the genotoxic *pks* island is more prevalent in patients with colorectal cancer. *Cougnoux et al. (2014)*, while not performing a direct replication, extended the findings of Arthur and colleagues to explore the mechanism of *pks*-produced colobactin toxicity effects on colorectal cancer. Finally, Arthur and colleagues have since published further work exploring in greater detail the genetic mechanisms behind the association of colorectal cancer with colitis and colonization by *E. coli* NC101 with or without the *pks* island, in which they demonstrate that colon inflammation itself has an influence on the expression of the genotoxic *pks* island (*Arthur et al., 2014*).

## Materials and methods

Unless otherwise noted, all protocol information was derived from the original paper, references from the original paper, or information obtained directly from the authors. An asterisk (*) indicates data or information provided by the Reproducibility Project: Cancer Biology core team. A hashtag (#) indicates information provided by the replicating lab.

### Protocol 1: comparing the growth curves of *E. coli* NC101 and *E. coli* NC101Δ*pks*

This protocol describes how to grow both *E. coli* NC101 and *E. coli* NC1010Δ*pks* to compare their growth curve, as seen in Supplemental figure 7.

### Sampling

- This experiment will be repeated a total of three times.
    a. Power calculations were not performed, as no significant difference was reported in the original study.

### Materials and reagents

- Reagents that differ from those used originally are indicated with an asterisk (*).

| Reagent | Type | Manufacturer | Catalog # | Comments |
|---|---|---|---|---|
| *E. coli* NC101 strain | Cells | Original authors | n/a | |
| *E. coli* NC101Δpks strain | Cells | Original authors | n/a | |
| Luria–Bertani (LB) broth* | Materials | Sigma | L3022 | Original unspecified |

### Procedure

1. Streak *E. coli* strains NC101 and NC101Δpks on agar plates and incubate at 37˚C overnight.
2. Inoculate 5 ml Luria–Bertani broth (LB broth) with a single colony of each *E. coli* strain picked from the freshly streaked plates and grow for ∼8 hr at 37˚C in a shaking incubator.
3. Inoculate *E. coli* strains at a 1:500 dilution in 150 ml LB broth and incubate at 37˚C overnight (12–16 hr) in a shaking incubator.
4. Inoculate *E. coli* strains at 1:500 dilution in 10 ml LB broth and measure the OD600 of the diluted cultures (timepoint 0). Collect the remaining *E. coli* for genomic DNA extraction in protocol 2.
5. Incubate at 37˚C in a shaking incubator.
    a. Measure the OD600 of cultures every 20–30 min until saturation phase has been reached for both cultures (∼8 hr).
    b. Lab will note instrument make, model and RPM.
6. Repeat independently two additional times.

### Deliverables

- Data to be collected:
    a. Raw values for OD600 measurements at each time point for NC101 and NC101Δpks.
    b. Semi-logarithmic graph of average OD600 (log) vs time (linear) for NC101 and NC101Δpks.

- Sample delivered for further analysis:
   a. Cultures from Step 3 for use in Protocol 2.

## Confirmatory analysis plan

- Statistical analysis of the replication data:
   a. Comparison of growth curve fit by nonlinear regression.

## Known differences from the original study

- The 37°C incubator used by the lab may not be the same make and model as used in the original study. The lab will note the make and model of the incubator they use.

## Provisions for quality control

All data obtained from the experiment—raw data, data analysis, control data and quality control data—will be made publicly available, either in the published manuscript or as an open access dataset available on the Open Science Framework (https://osf.io/y4tvd/).

- This protocol will confirm that loss of the *pks* genotoxic island does not affect the growth curve of *E. coli* NC1010Δ*pks* as compared to its parental strain, *E. coli* NC101.

## Protocol 2: PCR amplification and sequencing of polyketide synthase (*pks*) genotoxic island in *E. coli* NC101 and NC101Δpks

This protocol describes the PCR amplification of the 5′ and 3′ ends of the *pks* genotoxic island to confirm its presence in *E. coli* NC101 and its absence in *E. coli* NC101Δ*pks.* This is a quality control step to confirm the absence of the *pks* island in the *E. coli* NC101Δ*pks* strain.

### Sampling

- This experiment will be performed once.
   a. Power calculations are not necessary for this PCR screen.
- The experiment consists of two cohorts:
   a. Genomic DNA from *E. coli* NC101.
   b. Genomic DNA from *E. coli* NC101Δ*pks.*
   c. Each cohort has four PCR reactions run:
      - L1 + L2: detects the 5′ end of the *pks* island.
      - R1 + R2: detects the 3′ end of the *pks* island.
      - 16S F + 16S R: control gene for amplification.
      - ClbB-F + ClbB-R: also detects the colibactin gene.
         i. Additional reaction recommended by original authors.

### Materials and reagents

- Reagents that differ from those used originally are indicated with an *.

| Reagent | Type | Manufacturer | Catalog # | Comments |
|---|---|---|---|---|
| *pks* L1 | Primer | At replicating lab's discretion | n/a | Original synthesis provider unspecified |
| *pks* L2 | Primer | At replicating lab's discretion | n/a | Original synthesis provider unspecified |
| *pks* R1 | Primer | At replicating lab's discretion | n/a | Original synthesis provider unspecified |
| *pks* R2 | Primer | At replicating lab's discretion | n/a | Original synthesis provider unspecified |
| 16S F | Primer | At replicating lab's discretion | n/a | Original synthesis provider unspecified |
| 16S R | Primer | At replicating lab's discretion | n/a | Original synthesis provider unspecified |
| ClbB-F | Primer | At replicating lab's discretion | n/a | Recommended by the original authors |
| ClbB-R | Primer | At replicating lab's discretion | n/a | Recommended by the original authors |
| Agarose* | Reagent | Sigma | A9539 | Original unspecified |
| Ethidium bromide* | Reagent | Sigma | E1510 | Original unspecified |

## Procedure

1. Extract bacterial genomic DNA from the sample collected in Protocol 1.
   a. Lab will document their methodology for bacterial genomic DNA extraction.
   b. Lab will include quality control data such as $A_{260}/A_{280}$ ratios from quantifying DNA concentration.
2. Run the following PCR reactions:
   a. Experimental.

| Template | Forward primer | Reverse primer |
|---|---|---|
| *E. coli* NC101 gDNA | L1 | L2 |
| *E. coli* NC101Δ*pks* gDNA | L1 | L2 |
| Water (no DNA control) | L1 | L2 |
| *E. coli* NC101 gDNA | R1 | R2 |
| *E. coli* NC101Δ*pks* gDNA | R1 | R2 |
| Water (no DNA control) | R1 | R2 |
| *E. coli* NC101 gDNA | ClbB-F | ClbB-R |
| *E. coli* NC101Δ*pks* gDNA | ClbB-F | ClbB-R |
| Water (no DNA control) | ClbB-F | ClbB-R |

   b. Controls (additional control).

| Template | Forward primer | Reverse primer |
|---|---|---|
| *E. coli* NC101 gDNA | 16S F | 16S R |
| *E. coli* NC101Δ*pks* gDNA | 16S F | 16S R |
| Water (no DNA control) | 16S F | 16S R |

   c. Primers.

| Primer | Sequence | Expected band size | Comment |
|---|---|---|---|
| L1 | 5′-AAT CAA CCC AGC TGC AAA TC-3′ | 1824 bp | L1 + L2 detect the 3′ end of the *pks* |
| L2 | 5′-CAC CCC CAT CAT TAA AAA CG-3′ | | |
| R1 | 5′-AGC CGT ATC CTG CTC AAA AC-3′ | 1413 bp | R1 + R2 detect the 5′ end of the *pks* |
| R2 | 5′-TCG GTA TGT CCG GTT AAA GC-3′ | | |
| ClbB-F | 5′-GCG CAT CCT CAA GAG TAA ATA-3′ | 280 bp | |
| ClbB-R | 5′-GCG CTC TAT GCT CAT CAA CC-3′ | | |
| 16S F | 5′-GTG STG CAY GGY TGT CGT CA-3′ | | |
| 16S R | 5′-GTG STG CAY GGY TGT CGT CA-3′ | | |

d. Reaction set-up.

| 10× buffer | 5 µl |
|---|---|
| 5 µM dNTPs | 0.5 µl |
| 50 mM MgCl$_2$ | 1.5 µl |
| 5 µM F primer | 0.5 µl |
| 5 µM R primer | 0.5 µl |
| Invitrogen Taq polymerase | 0.5 µl |
| Bacterial genomic DNA | 2 µl |
| Water | Bring up to 50 µl |

   e. Cycling parameters.
      i. Denature at 95°C for 5 min.
      ii. 35 (up to 50) cycles of:
         ■ 95°C for 45 s.
         ■ 56°C for 45 s.
         ■ 72°C for 45 s.
      iii. 72°C for 10 min.
      iv. Hold at 4°C forever.
3. Run out PCR amplicons on a 1.5% agarose gel alongside a size marker. Visualize with ethidium bromide.
4. Sequence *pks* amplicons by Sanger automated DNA sequencing.
   a. BLAST sequencing results against the *E. coli* Colibactin synthesis cluster (AM229678.1).
5. Sequence 16S amplicons to confirm identity of bacterial strains (additional control).

## Deliverables

- Data to be collected:
   a. Full gel image of ethidium bromide stained gel showing PCR products for *pks*-L, *pks*-R and 16S amplicons.
   b. Sequencing chromatograms.
   c. BLAST comparison results.

## Confirmatory analysis plan

- Statistical analysis of the replication data:
   a. No statistical test required.
   b. Visually confirm presence of *pks* bands in *E. coli* NC101 and absence in *E. coli* NC101Δ*pks*.

## Known differences from the original study

- Lab will use in-house bacterial genomic DNA extraction protocol.
   a. Original was unspecified.

## Provisions for quality control

All data obtained from the experiment—raw data, data analysis, control data and quality control data—will be made publicly available, either in the published manuscript or as an open access dataset available on the Open Science Framework (https://osf.io/y4tvd/).

- We will sequence the 16S rRNA of the bacteria to confirm the identity of the strain. The sample purity (A$_{260}$/A$_{280}$ ratio) of the extracted DNA will be recorded.

## Protocol 3: mono-associate mice with *E. coli* NC101 or NC101Δpks and analyze intestinal tumorigenesis and inflammation

This protocol describes the inoculation of germ-free *Il10*$^{-/-}$ mice with *E. coli* as well as treatment with the carcinogen azoxymethane (AOM), as seen in Figure 4.

## Sampling

- This experiment will use at least 10 mice per group for a final power of 80–96%.
   a. See 'Power calculations' section for details.
- The experiment consists of the following cohorts:
   a. Germ-free *IL10*$^{-/-}$ mice treated with AOM and inoculated with *E. coli* NC101, harvested at 18 weeks post AOM.
      ■ n = 14.
         i. Expected survival is 75% (as seen in Supplemental figure 10), thus 14 mice are required to predict 10 will survive.

b. Germ-free *IL10*<sup>−/−</sup> mice treated with AOM and inoculated with *E. coli* NC101Δ*pks*, harvested at 18 weeks post AOM.
- *n* = 16.
  i. Expected survival is 62.5% (as seen in Supplemental figure 10), thus 16 mice are required to predict 10 will survive.

## Materials and reagents

- Reagents that differ from those used originally are indicated with an *.

| Reagent | Type | Manufacturer | Catalog # | Comments |
|---|---|---|---|---|
| IL10<sup>−/−</sup> mice | Mice | Original lab | n/a | |
| Wild-type mice | Mice | Original lab | n/a | |
| Azoxymethane (AOM) | reagent | Sigma | A5486 | |
| Formalin* | Reagent | Sigma | HT501128 | original unspecified |
| 0.1 mm zirconium beads and bead beater | Equipment | BioSpec Products | 1107900-101 | |
| DNeasy kit | Reagent | Qiagen | 69504 | |
| 16S F | Primer | At replicating lab's discretion | n/a | |
| 16S R | Primer | At replicating lab's discretion | n/a | |

## Procedure

### Notes

- Experiment should be conducted by experimenters blinded to genotype and treatment group.
- Azoxymethane (AOM) can show lot-to-lot variation in potency, and will lose potency and gain toxicity over time. In order to minimize these effects, a single lot of AOM will be used throughout the experiment. AOM will be aliquoted into 25 mg/ml aliquots and stored at −80°C. A fresh aliquot will be used each time AOM is needed to avoid repeated freeze-thaw cycles.
  1. Breed and raise *IL10*<sup>−/−</sup> mice in germ-free isolators.
  a. Use both male and female mice; house sexes separately.
  b. House mice 2–4 mice per cage.
  c. 12 hr/12 hr light/dark cycle.
  d. Diet is Purine Lab Diet 3500.
  2. At age 7–12 weeks initiate program to induce colitis/colorectal cancer.
  a. Maintain mice in gnotobiotic isolators.
  3. Randomly assign Il10<sup>−/−</sup> mice to two groups. As each mouse becomes eligible for induction of colitis/colorectal cancer, randomly assign to a treatment group using the adaptive randomization approach with the gender of the mice as the covariate that is assessed as mice are sequentially assigned to a particular treatment group. Assignment will aim for a similar distribution of genders in each cohort while also taking into account the pre-determined size of each treatment group.
  a. Group 1: *E. coli* NC101; *n* = 14.
  b. Group 2: *E. coli* NC101Δpks; *n* = 16.
  4. Colonize mice by oral gavage and rectal swabbing (dip sterile Q-tip in culture and swab anus) with log phase growth bactera:
  a. Use 200 µl of an overnight bacterial culture with a concentration of $2 \times 10^9$ CFU/ml.
    i. Record OD600 of culture used.
    ii. Perform serial dilution and plating of the culture used to swab for quantitative culturing. Record the actual CFUs of the culture used to colonize each mouse.
  b. *E. coli* NC101 or *E. coli* NC101Δpks for *Il10*<sup>−/−</sup> mice.
    i. Maintain in separate gnotobiotic isolators that will contain only the bacterium of interest throughout the study.
  c. After 4 weeks, confirm colonization by stool culture. Note: Steps i through vi are derived from *Uronis et al. (2011)*. Confirm bacterial strain by 16S RT-PCR:
    i. Collect up to 500 mg of fecal material.
      - Pellets can be stored in an eppendorf tube at −80°C until processing.
    ii. Resuspend in lysis buffer with 20 mg/ml lysozyme and incubate at 37°C for 30 min.
    iii. Add 10% SDS and 350 µg/ml Proteinase K for further lysis.
    iv. Homogenize samples with a bead beater and 0.1 mm zirconium beads.
    v. Extract DNA with a DNeasy kit.

 vi. Amplify the bacterial 16S ribosomal RNA gene.
- Forward primer: 5′-GTG STG CAY GGY TGT CGT CA-3′.
- Reverse primer: 5′-ACG TCR TCC MCA CCT TCC TC-3′.
- See Protocol 2, Procedure Step 2d and e for reaction and annealing conditions.

 vii. *Sequence amplicons and BLAST sequencing results to confirm colonization with *E. coli*.

 d. Quantify level of colonization by serial dilution culture:
 i. Collect at least 200 mg of fecal material.
 ii. Serially dilute 200 mg fecal material.
 iii. Plate dilutions on LB agar plates.
- Dilute enough to resolve single colony forming units (CFUs).

 iv. Calculate the total number of CFUs per 200 mg fecal matter.

5. At the same time as first stool culture, intraperitoneally inject with 10 mg/kg azoxymethane (AOM).
6. Repeat AOM injections every week for 5 more weeks (6 weeks total).
7. 18 weeks after last AOM injection, sacrifice mice.
 a. #Mice are anesthetized with isofluorane in a drop jar. Once respiration has ceased, the mice are exsanguinated.
 b. *Collect stool and quantify colonization as performed in Step 5b.
8. Macroscopically examine tumor formation:
 a. Remove colons from the cecum to the rectum, flush with PBS, and splay longitudinally.
 b. Blindly count tumors per mouse and measure tumor diameter macroscopically. Image colon and tumors.
9. Prepare tissue for histological analysis:
 a. Collect distal colon tissue samples.
 b. Swiss-roll colon tissue samples from the distal to the proximal end and fix overnight in 10% formalin.
10. Paraffin-embed tissues.
 a. #The replicating lab uses an automated embedding station:
 i. Samples are passed through a dehydration series consisting of 70%, 80%, 2 × 95% and 3 × 100% ethanol for 30 min each wash.
 ii. Samples are washed into xylene; first wash is 30 min, the second is an hour.
 iii. The samples are washed into 57°C Paraplast; four washes of 30 min each.
 iv. Samples are mounted in mold and allowed to cool and harden.
11. Cut 6 μm sections and mount on slides.
12. Stain with hematoxylin and eosin for histologic analysis.
 a. Lab will record H&E staining protocol used.
13. Blindly score for inflammation, dysplasia, and invasion (scoring should by performed by an expert animal histopathologist).
 a. Score mucosal inflammation (0–4) by the degree of lamina propria mononuclear cell (LPMNC) infiltration, crypt hyperplasia, goblet cell depletion, and architectural distortion.
 i. See Table 2 in http://www.ncbi.nlm.nih.gov/pmc/articles/PMC507509/pdf/980945.pdf.
 b. Score dysplasia as follows:
 i. 0 = no dysplasia.
 ii. 1 = mild dysplasia characterized by aberrant crypt foci (ACF), +0.5 for small gastrointestinal neoplasia (GIN), or multiple ACF.
 iii. 2 = moderate dysplasia with GIN, +0.5 for multiple occurrences or small adenoma.
 iv. 3 = severe or high grade dysplasia restricted to the mucosa.
 v. 3.5 = adenocarcinoma, invasion through the muscularis mucosa.
 vi. 4 = adenocarcinoma, full invasion through the submucosa and into or through the muscularis propria.
 c. Score invasion as follows:
 i. 0 = no invasion.
 ii. 1 = 5–10% involvement.
 iii. 2 = 10–15% involvement.
 iv. 3 = 25–50% involvement.
 v. 4 = >50% involvement.
- Multiply invasion score by 1 if invasion is through the muscularis mucosa, and by 2 if invasion is through the muscularis propria and serosa.

## Deliverables

- Data to be collected:
 a. Mouse records (gender used in each group, type of colonization procedure, health records, condition for early euthanasia, etc).
 b. $OD_{600}$ of overnight bacterial cultures used in colonization.
 c. (Compare to Supplemental figure 10): Raw data and Kaplan–Meier survival curve of mice for all conditions.
 d. Sequencing chromatograms and gel image of amplicons from stool sample confirming colonization with *E. coli*.
 e. (Compare to Figure 4D): Images, raw numbers and dot plot graph of macroscopic tumor number per mouse (multiplicity) at 18 weeks of colon tissue from mice for all conditions.
 f. (Compare to Figure 4F): Micrographs of H&E histology for each mouse at 18 weeks for all conditions.

 g. (Compare to Figure 4E): Raw numbers and dot plot graph of mean macroscopic tumor diameter in each mouse at 18 weeks of colon tissue from mice for all conditions.

 h. (Compare to Figure 4A): Raw numbers and dot plot graph of inflammation scores at 18 weeks of colon tissue from mice for all conditions.

 i. (Compare to Figure 4B): Raw numbers and dot plot graph of neoplasia scores at 18 weeks of colon tissue from mice for all conditions.

 j. (Compare to Figure 4C): Raw numbers and dot plot graph of invasion scores at 18 weeks of colon tissue from mice for all conditions.

 k. Counts of bacterial colonization from stool sample per mouse at 4 week time point and at sacrifice.

## Confirmatory analysis plan

- Statistical analysis of the replication data:
    a. (As seen in Supplemental figure 10): Compare survival of AOM-treated $Il10^{-/-}$ mice inoculated with *E. coli* NC101 relative to NC101$\Delta pks$.
        - Log-rank test (Mantel Cox).
    b. (As seen in Figure 4A, right panel): Compare mean inflammation scores of AOM-treated $Il10^{-/-}$ mice mono-associated with *E. coli* NC101 relative to NC101$\Delta pks$.
        - Unpaired two-tailed Student's *t*-test.
    c. (As seen in Figure 4B): Compare mean neoplasia score of AOM-treated $Il10^{-/-}$ mice mono-associated with *E. coli* NC101 relative to NC101$\Delta pks$.
        - Unpaired two-tailed Student's *t*-test.
    d. (As seen in Figure 4C): Compare mean invasion score of AOM-treated $Il10^{-/-}$ mice mono-associated with *E. coli* NC101 relative to NC101$\Delta$pks.
        - Unpaired two-tailed Student's *t*-test.
    e. (As seen in Figure 4D): Compare mean macroscopic tumor number (multiplicity) of AOM-treated $Il10^{-/-}$ mice mono-associated with *E. coli* NC101 relative to NC101$\Delta pks$.
        - Unpaired two-tailed Student's *t*-test.
    f. (As seen in Figure 4E): Mean macroscopic tumor diameter per mouse of AOM-treated $Il10^{-/-}$ mice mono-associated with *E. coli* NC101 relative to NC101$\Delta pks$.
        - Unpaired two-tailed Student's *t*-test.
    g. Compare mean number of CFUs per 200 mg stool pellet between *E. coli* NC1010-colonized and *E. coli* NC1010$\Delta pks$-colonized mice at 4 weeks post-inoculation and at sacrifice.
        - Two-way ANOVA.
- Meta-analysis of original and replication attempt effect sizes:
    a. This replication attempt will perform the statistical analysis listed above, compute the effects sizes, compare them against the reported effect size in the original paper and use a meta-analytic approach to combine the original and replication effects, which will be presented as a forest plot.

## Known differences from the original study

- The replication attempt will be restricted to the 18 week time point.
- The microscope used in the original lab was an Olympus CX41; the replicating lab will use an Olympus BX41.
- We will be using a different lot of azoxymethane that used by the original authors. AOM is known to have lot-to-lot variation in efficacy that may affect the absolute numbers of tumors generated per mouse.
- The original study collected data on 4 female and 8 male $IL10^{-/-}$ mice inoculated with *E. coli* NC101 and 8 male $IL10^{-/-}$ mice inoculated with *E. coli* NC101$\Delta$pks. While the gender of the mice in the replication is not currently known, the mice will be randomly assigned when they reach the age for inoculation with the aim of a similar gender distribution in each group. This will likely generate a different gender distribution than the original study.

## Provisions for quality control

All data obtained from the experiment—raw data, data analysis, control data and quality control data—will be made publicly available, either in the published manuscript or as an open access dataset available on the Open Science Framework (https://osf.io/y4tvd/).

- The experiment will be performed by a Science Exchange lab with expertise in germ free mouse studies.
- Experimenters will be blinded to the genotype and treatment group. Mice will be randomly assigned to treatment groups.

## Power calculations

### Protocol 1

- Not applicable.

## Protocol 2

- Not applicable.

## Protocol 3

### Summary of original data

Figure 4A:

| 4A: Inflammation score at 18 weeks | Mean | SD | N |
|---|---|---|---|
| *IL10*⁻/⁻ mice inoculated with *E. coli* NC101 | 4 | 0 | 9 |
| *IL10*⁻/⁻ mice inoculated with *E. coli* NC101delta*PKS* | 3.8 | 0.45 | 5 |

### Test family

- Two-tailed t-test, difference between two independent means, alpha error = 0.05.
    a. Sensitivity calculations were performed using G*power software (*Faul et al., 2007*).

### Power calculations

- Because the original data shows a non-significant effect, we will not be powering this replication to detect an effect. Based on the sample size of 10 mice per group derived from Figure 4C, with α of 0.05 we will be powered to 80% to detect a Cohen's *d* of 1.3249474.

### Summary of original data

- Note: Raw data values obtained from scatterplot with confirmation of accuracy from the authors.

| 4B: Neoplasia score at 18 weeks with AOM treatment | Mean | SEM | SD | N |
|---|---|---|---|---|
| *IL10*⁻/⁻ mice inoculated with *E. coli* NC101 | 4.44 | 0.18 | 0.53 | 9 |
| *IL10*⁻/⁻ mice inoculated with *E. coli* NC101Δ*pks* | 3.60 | 0.24 | 0.55 | 5 |

- Stdev was calculated using formula SD = SEM*(SQRT n).

### Test family

- Two-tailed t-test, difference between two independent means, alpha error = 0.05.
    a. Power calculations were performed for statistically significant effects reported in original study using G*power software (*Faul et al., 2007*).

### Power calculations

| Group 1 vs | Group 2 | Pooled SD | Effect size | A priori power | Group 1 sample size | Group 2 sample size |
|---|---|---|---|---|---|---|
| NC101 | NC101Δ*pks* | 0.534 | 1.573034 | 83.2%* | 8* | 8* |

*Based on the sample size required for Figure 4C, we will use 10 mice per group. This brings the a priori power to 91.3%.

### Summary of original data

- Note: Raw data values obtained from scatterplot with confirmation of accuracy from the authors.

| 4C: Invasion score at 18 weeks with AOM treatment | Mean | SEM | SD | N |
|---|---|---|---|---|
| *IL10*⁻/⁻ mice inoculated with *E. coli* NC101 | 3 | 0.63 | 1.9 | 9 |
| *IL10*⁻/⁻ mice inoculated with *E. coli* NC101Δ*pks* | 0.8 | 0.37 | 0.84 | 5 |

- Stdev was calculated using formula SD = SEM*(SQRT n).

## Test family

- Two-tailed t-test, difference between two independent means, alpha error = 0.05.
  a. Power calculations were performed for statistically significant effects reported in original study using G*power software (*Faul et al., 2007*).

## Power calculations

| Group 1 vs | Group 2 | Pooled SD | Effect size | A priori power | Group 1 sample size | Group 2 sample size |
|---|---|---|---|---|---|---|
| NC101 | NC101Δ*pks* | 4.410 | 1.814059* | 80.1% | 6* | 6* |

*Based on the sample size required for 4C, we will use 10 mice per group. This brings the a priori power to 96.9%.

## Summary of original data

- Note: Raw data values obtained from scatterplot with confirmation of accuracy from the authors.

| 4D: Tumor number per mouse at 18 weeks with AOM treatment | Mean | SEM | SD | N |
|---|---|---|---|---|
| *IL10*⁻/⁻ mice inoculated with *E. coli* NC101 | 10.6 | 1.8 | 5.3 | 9 |
| *IL10*⁻/⁻ mice inoculated with *E. coli* NC101Δ*pks* | 2.6 | 0.7 | 1.5 | 5 |

- Stdev was calculated using formula SD = SEM*(SQRT n).

## Test family

- Two-tailed t-test, difference between two independent means, alpha error = 0.05.
  a. Power calculations were performed for statistically significant effects reported in original study using G*power software.

## Power calculations

| Group 1 vs | Group 2 | Pooled SD | Effect size | A priori power | Group 1 sample size | Group 2 sample size |
|---|---|---|---|---|---|---|
| NC101 | NC101Δ*pks* | 1.625 | 1.353846 | 81.7% | 10 | 10 |

Figure 4E:

| 4E: Mean macroscopic tumor diameter at 18 weeks | Mean | SD | N |
|---|---|---|---|
| *IL10*⁻/⁻ mice inoculated with *E. coli* NC101 | 3.02 | 0.74 | 9 |
| *IL10*⁻/⁻ mice inoculated with *E. coli* NC101delta*PKS* | 3.66 | 1.59 | 5 |

### Test family

- Two-tailed t-test, difference between two independent means, alpha error = 0.05.
  a. Sensitivity calculations were performed using G*power software (*Faul et al., 2007*).

### Power calculations

- Because the original data shows a non-significant effect, we will not be powering this replication to detect an effect. Based on the sample size of 10 mice per group derived from Figure 4C, with α of 0.05 we will be powered to 80% to detect a Cohen's *d* of 1.3249474.

    Supplemental figure 10:

### Test family

- Two-tailed t-test, difference between two independent means, alpha error = 0.05.
  a. Sensitivity calculations were performed using G*power software (*Faul et al., 2007*).

### Power calculations

- Because the original data shows a non-significant effect, we will not be powering this replication to detect an effect. Based on the sample size of 10 mice per group derived from Figure 4C, with α of 0.05 we will be powered to 80% to detect a Cohen's *d* of 1.1453705.

## Acknowledgements

The Reproducibility Project: Cancer Biology core team would like to thank the original authors, in particular Janelle C Arthur and Christian Jobin, for generously sharing critical information as well as reagents to ensure the fidelity and quality of this replication attempt. We would also like to thanks the following companies for generously donating reagents to the Reproducibility Project: Cancer Biology; American Tissue Culture Collection (ATCC), BioLegend, Cell Signaling Technology, Charles River Laboratories, Corning Incorporated, DDC Medical, EMD Millipore, Harlan Laboratories, LI-COR Biosciences, Mirus Bio, Novus Biologicals, Sigma–Aldrich, and System Biosciences (SBI).

## Additional information

### Group author details

**Reproducibility Project: Cancer Biology**

Elizabeth Iorns: Science Exchange, Palo Alto, California; William Gunn: Mendeley, London, United Kingdom; Fraser Tan: Science Exchange, Palo Alto, California; Joelle Lomax: Science Exchange, Palo Alto, California; Timothy Errington: Center for Open Science, Charlottesville, Virginia

### Competing interests

KE: The Germ Free Laboratory is a Science Exchange associated lab. RP:CB: We disclose that EI, FT, and JL are employed by and hold shares in Science Exchange Inc. The experiments presented in this manuscript will be conducted by KE at the Germ Free Laboratory, which is a Science Exchange lab. The other authors declare that no competing interests exist.

### Funding

| Funder | Author |
| --- | --- |
| Laura and John Arnold Foundation | Reproducibility Project: Cancer Biology |

The Reproducibility Project: Cancer Biology is funded by the Laura and John Arnold Foundation, provided to the Center for Open Science in collaboration with Science Exchange. The funder had no role in study design or the decision to submit the work for publication.

## Author contributions

KE, WY, Drafting or revising the article; RP:CB, Conception and design, Drafting or revising the article

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
