## [Decision Letter]

Thank you for sending your work entitled “Registered report: Intestinal
inflammation targets cancer-inducing activity of the microbiota” for
consideration at *eLife*. Your Registered report has been favorably
evaluated by Richard Losick (Senior editor), a guest editor, 4 reviewers, and a
biostatistician.

The Reviewing editor and the other reviewers discussed their comments before we reached
this decision, and the Reviewing editor has assembled the following comments to help you
prepare a revised submission.

The proposed experiments are largely appropriately designed although there are important
points of clarification required.

1) The authors accurately summarize the literature directly related to the original
paper, but should also be aware of two key manuscripts relevant to Colibactin: PMIDs
20534522 (PNAS 107:11537, 2010) and 16902142 (Science 313:848, 2006).

2) There are key concerns regarding the azoxymethane lot and its handling and
documentation. First, the authors should note that each lot of azoxymethane may have
differing potency, so one lot should be used throughout this replication experiment.
Furthermore, this lot should first be tested for its ability to induce tumors in the
AOM/DSS model. Using a different lot of azoxymethane from the original study could
result in differences in tumorigenesis (i.e. penetrance, size, invasion, etc.) between
the original study and the reproduced study. Importantly though, this should not affect
differences in tumorigenesis between experimental groups but could lead to
uninterpretable results if tumorigenesis is very low in the *E. coli*
NC101, positive control group, associated with the lot of AOM. Second, the replication
authors should describe better how they aliquot, store and use AOM, which soon after
dilution becomes less stable and less carcinogenic, but more toxic and therefore can
cause unexpected mortality that would hamper the experimental design.

3) While this project has been under review, another original paper has been published
by the authors of the paper to be replicated: Arthur et al., Nature Communications, Sept
2014 (on-line). The *E. coli* NC101 tumorigenesis model is further
explored in this manuscript. In consultation with the original authors, the replication
project should clarify the contribution of AOM to the model. In Uronis et al. (PLOS ONE
4:e6026, 2009), it appears that *IL10*^*-/-*^
mice alone were not evaluated for tumorigenesis upon colonization with *E.
coli* NC101. In this replication project, the text of the original Science
paper suggests that *IL10*^*-/-*^ is insufficient
to promote tumorigenesis upon colonization with NC101 whereas in the most recent Arthur
publication noted above, *IL10*^*-/-*^ mice
exhibit similar tumorigenesis, but not invasive extent of neoplasia, compared to the AOM
+ *IL10*^*-/-*^ model (Figure 4, Nature
Communications, on-line as above). The authors of the replication project should comment
on the impact of these recent data on their replication plan. We query whether the
replication authors should consider including IL-10^-/-^ mice colonized with
*E. coli* NC101 without AOM treatment as a group into their
experiments.

4) There is no explicit indication that the mice in the gnotobiotic study will be
age-matched or sex-matched. It is only stated that cancer/colitis will be induced at
7-12 weeks and that male and female mice will be used. Although not explicitly stated,
one assumes that male and female mice will be housed in separate isolators. Will the
numbers of male and female mice used approximate that in the original experiments?

5) Because application of histopathologic criteria can be somewhat subjective and
pathologist-dependent, the authors should consider whether histologic images should be
evaluated independently by two pathologists and, if possible, by the animal
histopathologist who evaluated the images in the original publication. The scoring
system is specific about lamina propria mononuclear cell infiltration but
polymorphonuclear infiltration should also be considered.

6) The method of inoculation allows for more than one approach although the original
paper utilized gavage and rectal swabbing. While the authors suggest colonization with
oral swab as an alternative, oral gavage is likely to yield more reproducible
colonization results. Please clarify.

7) Please specify the typical OD600 or range considered acceptable to be used for the
bacterial inoculation cultures. Confirmation of exact CFUs and culture purity by
quantitative culturing of the inoculum should be performed. Confirmation that the
inoculated strains are ClbB-positive and ClbB-negative should be done. Similarly,
cultures to verify colonization with the specific strain inoculated at week 4 and at the
end of the experiment, should be completed.

8) Sampling. It is not clear why the expected mortality for *E. coli*
NC101Δpks is higher, when this strain induces less tumorigenesis than the parent
strain *E. coli* NC101.

9) Can the authors clarify the approach/plan if the proposed experiments do not
replicate the original data? Is there any back-up plan to adjust the experimental plan
to adjust for potential technical concerns such as the potency of the azozymethane lot
or if the experimental course differs in a different facility (e.g., more mouse
death)?

Statistical comments:

1) Cross-study variation should be taken into account to determine power pre-data
collection. This is hard to estimate, but papers by Giovanni Parmigiani and
collaborators at the Dana Farber provide some estimates about cross-study variation that
could be used for this purpose. The authors should budget some additional variability
because of cross-study reproducibility, and increase the sample size on-the-fly, as they
deem appropriate.

2) The final report on the replicated study should report the actual power of the tests,
based on the standard deviations in the replicated study.

3) Please clarify the comment 'based on the sample size of 13 mice per group
derived from Figure 4C'. The sample size for Figure 4C is stated as 10 to achieve
80% power, α = 0.05.

[Editors' note: further revisions were requested prior to acceptance, as
described below.]

Thank you for resubmitting your article entitled “Registered report: Intestinal
inflammation targets cancer-inducing activity of the microbiota” for further
consideration at *eLife*. Your revised Registered Report has been
favorably evaluated by Richard Losick (Senior editor), a guest Reviewing editor, three
of the original reviewers, and a biostatistician.

There are two remaining issues that need to be addressed before acceptance, as outlined
below:

1) Response to point 7: the reviewers feel strongly that quantitative cultures of the
inoculum to prove purity and numbers of organisms inoculated should be performed.
Relying on OD600 solely does not account for culture purity, nor potentially nonviable
organisms in the culture.

2) Response to point 1 of the statistical review: one approach to incorporate putative
extra variation in the proposed study would be to calculate sample size by assuming a
range of the anticipated effect size. For example, the effect size for, say, Figure 4C
used in the power calculation is 1.353846. So, one may examine sample size for smaller
effect size: say, effect size of 1.2 and 1.1 and see how much the sample size would
change in these two additional settings. When we do this calculation, we can see that
the sample size is approximately 11 mice in each group if the effect size is 1.2, and
approximately 13 mice in each group when the effect size is 1.1. The sample size does
not increase drastically under these additional effect sizes. So, perhaps, the sample
size of 10 is fine. Of course, if the observed effect size at the end of the study is
less than 1.353846, then the result is likely going to be non-significant since the
study was powered to detect an effect size of 1.353846 or more.

We would not recommend a post-hoc power calculation. But the investigators should
consider saying upfront in the Methods section (when they publish their work) that they
calculated the power at the beginning of the study using a two-sample t-test framework
to detect a certain effect size.

---

## [Author Response]

*1) The authors accurately summarize the literature directly related to the
original paper, but should also be aware of two key manuscripts relevant to
Colibactin: PMIDs 20534522 (PNAS 107:11537, 2010) and 16902142 (Science 313:848,
2006)*.

Thank you for bringing these two papers to our attention. We have added references to
these papers in the Introduction.

*2) There are key concerns regarding the azoxymethane lot and its handling and
documentation. First, the authors should note that each lot of azoxymethane may have
differing potency, so one lot should be used throughout this replication experiment.
Furthermore, this lot should first be tested for its ability to induce tumors in the
AOM/DSS model. Using a different lot of azoxymethane from the original study could
result in differences in tumorigenesis (i.e. penetrance, size, invasion, etc.)
between the original study and the reproduced study. Importantly though, this should
not affect differences in tumorigenesis between experimental groups but could lead to
uninterpretable results if tumorigenesis is very low in the* E. coli
*NC101, positive control group, associated with the lot of AOM. Second, the
replication authors should describe better how they aliquot, store and use AOM, which
soon after dilution becomes less stable and less carcinogenic, but more toxic and
therefore can cause unexpected mortality that would hamper the experimental
design*.

We would like to thank the reviewers for bringing these details to our attention.
Unfortunately it is beyond the scope of this project to include a full pilot study
testing the lot of AIM independently in addition to the experimental arms already
proposed. We will clearly note, however, in the Replication Study that details the
results of the replication experiment that lot-to-lot variability in the effectiveness
of azoxymethane needs to be taken into consideration when comparing the original and
replication data. When we are performing the comparative analysis of the replication and
original data, we will assess both the quantitative numbers of tumors (which may be
affected by differences in AOM lots), but also the trends between experimental groups,
which should not be. Our replicating lab, which has previous experience working with
AOM, will make a 25 mg/mL stock solution of AOM and store it in aliquots at
-80°C. They will use a fresh aliquot each time they need AOM to prevent multiple
freeze thaw cycles. Additionally, we will note in the “Known differences from the
original study” section that we will be using a different lot of AOM, and that
AOM is known to have lot-to-lot variation in efficacy. The manuscript has been updated
to reflect this change.

*3) While this project has been under review, another original paper has been
published by the authors of the paper to be replicated: Arthur et al., Nature
Communications, Sept 2014 (on-line). The* E coli *NC101 tumorigenesis
model is further explored in this manuscript. In consultation with the original
authors, the replication project should clarify the contribution of AOM to the model.
In Uronis et al. (PLOS ONE 4:e6026, 2009), it appears that*
IL10^-/-^
*mice alone were not evaluated for tumorigenesis upon colonization with E coli
NC101. In this replication project, the text of the original Science paper suggests
that* IL10^-/-^
*is insufficient to promote tumorigenesis upon colonization with E coli NC101
whereas in the most recent Arthur publication noted above,*
IL10^-/-^
*mice exhibit similar tumorigenesis, but not invasive extent of neoplasia,
compared to the AOM +* IL10^-/-^
*model (Figure 4, Nature Communications, on-line as above). The authors of the
replication project should comment on the impact of these recent data on their
replication plan. We query whether the replication authors should consider including
IL-10*^*-/-*^
*mice colonized with* E. coli *NC101 without AOM treatment as a
group into their experiments*.

We thank the reviewers for bringing this additional literature to our attention. From a
purely intellectual perspective it is interesting to consider the contributions of each
factor—AOM, genetic status and colonization with types of bacteria—to
developing colorectal cancer. However, the aim of the Reproducibility Project: Cancer
Biology is to replicate as faithfully as possible experiments, and not the conclusions
and implications drawn from those experiments. As such, it is beyond the scope of this
project to add an additional arm of mouse treatments that was not part of the original
experiment based on data from a separate paper. We have included a discussion of the
Arthur 2014 paper in the Introduction to describe the relevance of these newer findings
to our replication study.

*4) There is no explicit indication that the mice in the gnotobiotic study will
be age-matched or sex-matched. It is only stated that cancer/colitis will be induced
at 7-12 weeks and that male and female mice will be used. Although not explicitly
stated, one assumes that male and female mice will be housed in separate
isolators*. *Will the numbers of male and female mice used approximate
that in the original experiments?*

Our intention is indeed to use similar numbers of male and female mice as used in the
original experiment. As of the writing of this letter, we are currently negotiating the
MTA with the UNC Gnotobiotic core for our lab to receive the IL10 null mice. Due to the
efforts involved in breeding and treating these mice, the replicating lab will
approximate as closely as possible the same numbers of male and female mice used, based
on the sexes they obtain in their breeding. We will record the sex of each mouse for
later comparison to the original numbers of male and female mice used. The manuscript
has been updated to include this detail.

5) Because application of histopathologic criteria can be somewhat subjective
and pathologist-dependent, the authors should consider whether histologic images
should be evaluated independently by two pathologists and, if possible, by the animal
histopathologist who evaluated the images in the original publication. The scoring
system is specific about lamina propria mononuclear cell infiltration but
polymorphonuclear infiltration should also be considered.

This is a strong point to consider. However, the aim of the Reproducibility Project:
Cancer Biology is to perform independent replications, and to assess if an effect is
repeatable when performed by a separate group of people in a separate lab. Given that
the person who scored the slides was Dr. Janelle Arthur, the original study’s
first author (along with a second qualified colleague), we feel that involving her in
the scoring of the replication data would affect the independence of the replication. To
address the subjective nature of the histopathological scoring, Dr. Arthur has kindly
provided detailed guidelines to the histopathologist who will score the replication
data. In addition, the raw images of the stained tissues will be made publicly available
through the Open Science Framework, allowing the public to see and evaluate the data
directly and not be solely reliant upon each image’s score. This will allow
examination of features of the stained tissues not included in the original criteria for
scoring, such as polymorphonuclear infiltration.

*6) The method of inoculation allows for more than one approach although the
original paper utilized gavage and rectal swabbing. While the authors suggest
colonization with oral swab as an alternative, oral gavage is likely to yield more
reproducible colonization results. Please clarify*.

Thank you for clarifying the potential differences between the two methods. We will use
oral gavage to colonize the mice. We have updated the manuscript to reflect this
change.

*7) Please specify the typical OD600 or range considered acceptable to be used
for the bacterial inoculation cultures. Confirmation of exact CFUs and culture purity
by quantitative culturing of the inoculum should be performed. Confirmation that the
inoculated strains are ClbB-positive and ClbB-negative should be done. Similarly,
cultures to verify colonization with the specific strain inoculated at week 4 and at
the end of the experiment, should be completed*.

The *E. coli* strains being used in this project have been shared
directly by the original authors. We will verify them by PCR for the
*pks* genotoxic island, as outlined in Protocol 2. Steps to confirm
colonization of the mice by stool culture are already present in the protocol. Please
see Protocol 3 Step 4d for confirmation of colonization 4 weeks after inoculation and
step 7b for the second confirmation of colonization at the end of the study. Based on
direct communication from the first author, we are not aware of any exclusion criteria
based on an OD600 reading for using an overnight culture to inoculate mice. We will be
recording the OD600 of each culture used (see Protocol 3 Step 4a).

*8) Sampling. It is not clear why the expected mortality for* E. coli
*NC101Δpks is higher, when this strain induces less tumorigenesis than
the parent strain* E. coli *NC101*.

Percent survival was calculated from the Kaplan-Meier curve in Supplemental Figure 10 of
the original paper. The difference in percent survival was found by the original authors
to not be statistically significant; however we wished to ensure that enough mice would
survive the experiment in order to have highly powered statistical analysis.

9) Can the authors clarify the approach/plan if the proposed experiments do not
replicate the original data? Is there any back-up plan to adjust the experimental
plan to adjust for potential technical concerns such as the potency of the
azozymethane lot or if the experimental course differs in a different facility (e.g.,
more mouse death)?

The overall aim of the Reproducibility Project: Cancer Biology is in fact to attempt to
evaluate the replicability of key experiments from this paper. It will be an interesting
result if the experiments do or do not replicate. There are many reasons besides
inaccuracy of the original experiment that could result in non-replication; the
reviewers themselves have highlighted some already. The Replication Study that will be
published upon completion of data collection will include a thorough discussion of
reasons the experiment could have succeeded or failed to replicate, including variation
in the efficacy of AOM. Another point the reviewers explore is if we happen to have more
deaths amongst our mice than was seen by the original authors. Because of the scope of
this project, we would not be able to add additional mice in the event of experimental
animals dying, which is why we have attempted to estimate a sufficient number of animals
to ensure a fully powered dataset is achieved at the end of the experiment. However,
even if unexpected deaths do reduce the number of animals in the replication dataset
resulting in the replication being underpowered, we still believe there will be value in
assessing the effect size determined from the remaining animals.

*Statistical comments*:

*1) Cross-study variation should be taken into account to determine power
pre-data collection. This is hard to estimate, but papers by Giovanni Parmigiani and
collaborators at the Dana Farber provide some estimates about cross-study variation
that could be used for this purpose. The authors should budget some additional
variability because of cross-study reproducibility, and increase the sample size
on-the-fly, as they deem appropriate*.

We thank the reviewers for these suggestions. The cross-study variation, such as
approaches that utilize the 95% confidence interval of the effect size, can be useful in
conducting power calculations when planning adequate sample sizes for detecting the true
population effect size, which requires a range of possible observed effect sizes.
However, the Reproducibility Project: Cancer Biology is designed to conduct replications
that have 80% power to detect the point estimate of the originally reported effect size.
While this has the limitation of being underpowered to detect smaller effects than what
is originally reported, this standardizes the approach across all studies to be designed
to detect the originally reported effect size with at least 80% power. Also, while the
minimum power guarantee is beneficial for observing a range of possible effect sizes,
the experiments in this replication, and all experiments in the project, are designed to
detect the originally reported effect size with a minimum power of 80%. Thus, performing
power calculations during or after data collection is not necessary in this replication
attempt as all studies included are already designed to meet a minimum power or are
identified beforehand as being underpowered and thus are not included in the
confirmatory analysis plan. The papers by Giovanni Parmigiani and collaborators
highlight the importance of accounting for variability that can occur across different
studies, specifically gene expression data. While it is possible for a difference in
variance between the originally reported results and the replication data, this will be
reflected in the presentation of the data and a possible reason for obtaining a
different effect size estimate.

*2) The final report on the replicated study should report the actual power of
the tests, based on the standard deviations in the replicated study*.

We do not see the value in performing post-hoc power calculations on the obtained data.
However, we do agree that reporting the actual power of the tests to detect the
originally reported effect size estimate based on the sample size analyzed in the
replication study is important and will be reported.

3) Please clarify the comment 'based on the sample size of 13 mice per
group derived from Figure 4C'. The sample size for Figure 4C is stated as 10
to achieve 80% power, α = 0.05.

Thank you for identifying this typo. We have corrected it in the manuscript.

[Editors' note: further revisions were requested prior to acceptance, as
described below.]

*There are two remaining issues that need to be addressed before acceptance, as
outlined below*:

*1) Response to point 7: the reviewers feel strongly that quantitative cultures
of the inoculum to prove purity and numbers of organisms inoculated should be
performed. Relying on OD600 solely does not account for culture purity, nor
potentially nonviable organisms in the culture*.

We agree with the reviewers that quantitative culturing is an important step. Based on
our understanding, as well as that of the experts at the Germ Free Laboratory, the
procedures we propose satisfy the terms of quantitative culturing to determine the exact
CFUs of bacteria in the inoculum and also confirm living organisms were inoculated. We
have included a letter from Dr. Kate Eaton, the director of the Germ Free Laboratory,
detailing their standard procedures for quantitative culturing of inoculum and
confirmation of colonization.

*2) Response to point 1 of the statistical review: one approach to incorporate
putative extra variation in the proposed study would be to calculate sample size by
assuming a range of the anticipated effect size. For example, the effect size for,
say, Figure 4C used in the power calculation is 1.353846. So, one may examine sample
size for smaller effect size: say, effect size of 1.2 and 1.1 and see how much the
sample size would change in these two additional settings. When we do this
calculation, we can see that the sample size is approximately 11 mice in each group
if the effect size is 1.2, and approximately 13 mice in each group when the effect
size is 1.1. The sample size does not increase drastically under these additional
effect sizes. So, perhaps, the sample size of 10 is fine. Of course, if the observed
effect size at the end of the study is less than 1.353846, then the result is likely
going to be non-significant since the study was powered to detect an effect size of
1.353846 or more*.

*We would not recommend a post-hoc power calculation. But the investigators
should consider saying upfront in the Methods section (when they publish their work)
that they calculated the power at the beginning of the study using a two-sample
t-test framework to detect a certain effect size*.

We agree with the reviewers and plan to be fully transparent concerning our choice of
statistical analyses, both in the Registered Report as well as the follow-up Replication
Study. Additional materials relating to the Registered Report, including details on
power calculations, are posted on the study’s wiki page on the Open Science
Framework, and are made public upon publication of the Registered Report.